# High-Sensitivity CMOS-Integrated Floating Gate-Based UVC Sensors

**DOI:** 10.3390/s23052509

**Published:** 2023-02-24

**Authors:** Michael Yampolsky, Evgeny Pikhay, Ruth Shima Edelstein, Yakov Roizin

**Affiliations:** Tower Semiconductors, Migdal HaEmek 2310502, Israel

**Keywords:** UV sensors, sterilization, CMOS, floating gate

## Abstract

We report on novel UVC sensors based on the floating gate (FG) discharge principle. The device operation is similar to that of EPROM non-volatile memories UV erasure, but the sensitivity to ultraviolet light is strongly increased by using single polysilicon devices of special design with low FG capacitance and long gate periphery (grilled cells). The devices were integrated without additional masks into a standard CMOS process flow featuring a UV-transparent back end. Low-cost integrated UVC solar blind sensors were optimized for implementation in UVC sterilization systems, where they provided feedback on the radiation dose sufficient for disinfection. Doses of ~10 µJ/cm^2^ at 220 nm could be measured in less than a second. The device can be reprogrammed up to 10,000 times and used to control ~10–50 mJ/cm^2^ UVC radiation doses typically employed for surface or air disinfection. Demonstrators of integrated solutions comprising UV sources, sensors, logics, and communication means were fabricated. Compared with the existing silicon-based UVC sensing devices, no degradation effects that limit the targeted applications were observed. Other applications of the developed sensors, such as UVC imaging, are also discussed.

## 1. Introduction

UV sensing has been widely researched in view of numerous applications, such as sterilization, flame monitoring, UV spectroscopy, UV cure processes, measuring of solar indexes, UV communications, etc. [1,2,3,4,5,6]. Sterilization by UV radiation is extensively used in the past decades. UVC (200 nm to 280 nm) and UVB (280 nm to 320 nm) radiation are very efficient in eliminating dangerous microbes in air, water, and on different surfaces. The COVID-19 pandemic has sparked additional interest in using UV light for disinfection. It was reconfirmed by several groups that ultraviolet efficiently kills viruses, including SARS-CoV-2 [3,4,5,6]. UVB and UVC are destructive for COVID-19 RNA, while UVC of wavelength shorter than 260 nm also damages the protein coat of COVID-19 viruses. The reported doses necessary for efficient disinfection are the range of tens of mJ/cm^2^ (or even 3.7 mJ/cm^2^ [3]) depending on the wavelength, properties of sterilized surfaces and ambient conditions [3,4,5]. The sterilization of COVID viruses strongly stimulated the production of germicidal mercury lamps, UVC LEDs and various UV irradiating systems. UV sensors are needed to guarantee that the processed surfaces receive sufficient doses for sterilization and ensure the safety of people in rooms, cars, airplane cabins and other places where irradiation is performed. To guarantee the efficiency of sanitizing, it is necessary to measure the dose of ultraviolet light in multiple locations and in the presence of intense visible and IR illumination. Often, it is desirable to use UV sensors as elements of wireless sensor networks. Low-cost integrated solutions comprising sensors and low-power communication circuits, such as RFID and Wi-Fi connectivity, are required for such systems. There are several types of UV sensors embedded into CMOS, which are mostly based on specially designed photodiodes [2,7,8,9]. Corresponding devices assume additional masks in addition to the core CMOS process flow and typically have low responsivity in the UVC/UVB range.

One type of sensors successfully integrated into a standard CMOS process flow (bulk silicon without additional masks) is based on the FG (floating gate) discharge principle [10,11,12,13,14]. Those devices have the advantage of lower power consumption and a smaller footprint compared with photodiode-based solutions. UV radiation results in electron photoemission from the FG, changing the threshold voltage of MOS transistors comprising or coupled to the irradiated FG. This allows for precise dose measurement, without a special photo-current-integrating circuit employed in diode-based sensors. The FG sensors do not suffer from the effects of temperature-dependent dark currents that limit applications of diode sensors. Moreover, FG sensors are solar blind, due to the energy barrier for electrons between the FG and the surrounding silicon dioxide. The mechanism of discharge is discussed in detail in Section 2. C-Flash devices (a version of NVM cells developed at Tower Semiconductor) were used as the basis for designing the FG UVC sensors. In this paper, we report on the peculiarities of the developed devices and fabrication technology and provide the results of their characterization.

A demonstrator of a sterilization system, consisting of a UVC source, reader and multiple edge-sensing units, was built using the developed sensors and was presented as an example of possible applications.

Another considered application is UVC imaging. For this purpose, the FG UV sensors were integrated into a matrix read-out chip comprising 64 k of UVC-sensitive floating gates.

## 2. Materials and Methods

### 2.1. Standard CMOS Platform with Integrated FG UV Sensors

In [15], we reported UV sensors in the SOI with thin device layers. This solution employed strings of lateral p-i-n diodes as UV sensing elements. Compared with SOI sensors, FG devices assume relatively long (of the order of seconds) measurement times but can be realized on a standard bulk CMOS platform. Standard CMOS platforms allow fabricating much cheaper devices compared with SOI and typically have a wider selection of standard cells and IPs, which is important for SoC designs. The C-Flash NVM cell (example shown in Figure 1) was verified in several standard CMOS processes (180 nm and 65 nm technology nodes) without adding extra layers or process steps.

The C-Flash cells consist of two polysilicon capacitors: Control Gate (CG), Tunneling Gate (TG) for programming and erasing of the cells, and a read transistor coupled to the common floating gate. A technology with 110 Å GOX was selected for the UVC sensors exemplary designs, since it allowed increased reliability (absence of SILC leakage channels) after sensors’ multiple use (“cycling” in the NVM language).

Like all single polysilicon-based EPROMs with partially transparent to UVC back-end dielectrics, C-Flash can be erased by ultraviolet light. Nevertheless, the sensitivity is low if the device is supposed to work as a UV sensor. Typical erase times with UVC EPROM erase lamps are of the order of tens of minutes, and the needed energy is ~10–20 J/cm^2^. 

The FG made of ~2500 Å thick polysilicon (Poly) absorbs UVC photons (absorption depth < 100 Å), so that most of the generated photoelectrons do not reach the poly-SiO_2_ surface. The photoionized electrons which are transferred over the energy barrier into the silicon dioxide (and then to the substrate) are generated only at the edges of the polysilicon floating gate adjacent to the shallow trench isolation (STI), as shown in Figure 2b. Hence, the discharge is more efficient in devices with a long FG periphery. Strong improvement of the cell sensitivity to UV was obtained by using a “grilled” shape of the Polysilicon CG, thus increasing the periphery, as shown in Figure 2a. This improved cell is named Grilled C-Sensor in this paper. Additionally, the FG was built on the top of a 0.3 µm thick STI layer. This allowed decreasing the capacitance of the cell for a given area occupied by the FG and thus makes the measured threshold voltage *V_t_* more sensitive to small changes of the FG charge Δ*Q* (since Δ*Q* = *C* · Δ*V_t_*).

The FG is charged using a tunnel injection of electrons by applying the voltage between TG and CG terminals (Figure 2b) and discharged by UVC photons. The discharge of FG happens mainly by the excitation of electrons from the valance band of the charged poly floating gate over the energy barrier at the poly-SiO_2_ interface. The barrier is ~4.3 eV, as shown in Figure 2c. Energy of 4.3 eV corresponds to the wavelength of ~288 nm. This makes the FG sensor solar blind, i.e., not sensitive to wavelengths in the visible range. Compared with standard FG NVM, the discharge rate is strongly enhanced due to the low capacitance and long periphery of the floating gate. Sensitivity practical for UVC disinfection applications is thus achieved. The device can be charged and discharged repeatedly up to 10,000 times and has no significant sensitivity degradation after cycling and exposure to UV, as shown in Figure 3 (compared, e.g., with the sensors presented in [16]).

In light of the device capability to register the changes of the FG charge, it can be used to measure the incident UV dose without the need for an integrator. Additionally, because the sensor is being charged before the measurement and then does not require a voltage source during exposure to UV, it consumes no static power. These features allow creating very low power and small area UV dose measurement sensors.

### 2.2. CMOS Back-End Development

The back end of the UVC sensor devices comprises a four-level metal integration, with dielectric interlayers, as in the standard 0.18 µm technology platform. Modifications have been made to exclude UV absorbing layers and interference. The work was performed in two stages. At first, all incidences of absorbing layers within the back end were mapped. Typically, these are layers or stacks of layers containing SixNy. For simplicity, a one-level metal flow containing an FG memory device (Y-Flash) was used for the study of UV-facilitated discharge of various FG memories. Different nitrides or oxide/nitride stacks were deposited on top of the metal, and contacts to pads were enabled. Table 1 lists the stacks evaluated. Layers 1–5 are combinations of TEOS-based oxides with various types of SixNy, which are all deposited by PECVD. Layer 6 has been specifically engineered to have low compressive stress, low hydrogen content and high UV transmittance. It has been extensively characterized for its deposition rate, film uniformity, refractive index and wet etch rate. A sample without any dielectric deposited on the metal was used as a reference. The memory cells were charged to 4 V, and Vt was monitored during the exposure to different UV LEDs. 

The experiments allowed to determine which layers are significantly absorbing UV radiation, and hence need to be changed in integration, and which layers are good candidates as a replacement. Large Vt decay under UV radiation indicates that UV radiation is not absorbed or reflected by the interdielectric or passivation layers above it. SixNy layers in the stacks are typically thin <1000 Å. Hence, their impact on the total UV absorption is much less significant than that of passivation layers, which are 3000–6000 Å thick. As shown in Figure 4, the reference device named “no layer” exhibits the steepest decay. Interdielectric layers 2–3 exhibit a decay almost as steep, while interdielectric layer 1 exhibited a small decay. When looking at passivation layer alternatives, layers 1–2 exhibit very small decay of Vt under UV radiation. This is not surprising in light of the much thicker SixNy layers within those stacks. Layer 3 decay, on the other hand, is almost as steep as for the reference sample (no layer) and hence was adopted as the layer of choice for the UV sensor passivation. 

In the second stage of development, the chosen interdielectric layers and the passivation layer were implemented in a standard 4-level 0.18 µm technology node process flow. Various parameters of sensors with optimized back-end dielectrics were investigated, such as fill factor, discharge rate upon UV illumination with different wavelengths, etc.

### 2.3. Integrating the Sensors into a Measurement System

The discussed FG UV sensors were integrated into a system consisting of multiple edge units (Figure 5a), a UVC source and a reader (as described in Figure 5b). We were able to register the UVC doses and assess whether it was sufficient for disinfection on different surfaces. Since the FG UV sensor can be integrated with standard CMOS components on one chip, the edge units could be designed to be self-powered (e.g., by UV illumination) with a built-in communication module.

Another use for the FG UV sensors is UVC imaging, which was completed by building a matrix of 64 k FG cells integrated with a read-out circuitry.

## 3. Results and Discussion

### Performance of FG UV Sensor in a Standard CMOS Platform

The main parameters that influence the performance of the C-Sensors are the poly (FG) area and periphery, because as mentioned above, the sensitivity is higher if the sensing area is larger for the same capacitance.

Several types of FG devices were compared, as shown in Figure 6: (i) C-Flash: standard memory cell; (ii) Grill C-Flash: cell with modified FG for larger periphery; (iii) C-Sensor: cell with FG on STI for smaller capacitance; (iv) Grill C-Sensor: cell with all previous modification for both larger periphery and smaller capacitance.

Figure 7 shows the sensitivity comparison of the four FG device types. All the cells were programmed to a starting threshold voltage (Vt) of 4 V and then exposed to 255 nm UV light in steps of 10 seconds up to one minute. The “grilled” structures and C-Sensor (FG on STI, but the poly Si area is not “grilled”) show an enhanced discharge rate of FG and hence higher sensitivity.

Discharging of the “grilled” C-Sensor (Figure 8) by UV with different wavelengths in the range 220–400 nm shows that the highest sensitivity is in the UVC (255 nm) range, while the sensitivity drops sharply when the wavelength is increased to the UVB and UVA range, which is consistent with the discharge model in Section 2. The devices did not show any response to visible and IR light. A certain decrease at sensitivity at 220 nm is attributed to higher reflection since the back end was optimized at 255 nm. With Vt measurement resolution of 2 mV, it is possible to measure UV doses as low as 10 µJ/cm^2^ for 1 second. EPROM devices featuring the same discharge mechanism and using a UV-transparent back end, but different geometry of the FG and thickness of oxide under it, are typically discharged in ~20 min (see graph for C-Flash NVM at Figure 7), which is ~50 times slower than for the suggested sensor.

Figure 9 shows the influence of poly Si grill fill factor (% of stripes area from the total area) on the sensitivity of the devices. A lower fill factor results in a larger FG periphery and smaller capacitance.

Smaller fill factor resulted in the maximum sensitivity at all UV wavelengths. The effect relates to lower total capacitance of the FG and the increase in the FG periphery length.

Figure 10 illustrates the feasibility of UV imaging using the developed FG sensors. The Vt change of pixels in the 7 × 7 mm matrix comprising FG C-sensors (256 rows × 256 columns) is shown after exposure with UV LED (280 nm) through a TS (stands for “Tower Semiconductor”) template (Figure 10a). The pixels were pre-charged to a reference Vt before the exposure (uniform charging of floating gates); then, the difference after exposure was measured. Read-out of the sensor array (Figure 10c) is completed by ramping the gate (CG) voltage on rows of FG cells and comparing them to rows of reference cells (NMOS transistors) with a reference voltage at the gate.

The difference of Vt (Figure 10b) after charging of the cells and exposure to 280 nm UV LED is ~0.5 V.

The advantages of our FG devices over other types of UVC sensors are summarized in Table 2. FG devices show excellent sensitivity, low power consumption, low cost and low degradation. The major limitation relates to the response time. Thus, the developed low cost integrated into CMOS sensors is recommended for applications that require the measuring of total doses of radiation.

## 4. Summary

Original FG UVC sensors were optimized and integrated as elements of a CMOS mass production process flow without additional masks. To increase the sensitivity, dielectrics transparent to UVC and their stacks were developed and implemented in the back end of the CMOS process. The developed embedded technology allows system-on-chip (SoC) designs where UV sensors and electronics for preliminary signal processing and RF communication are fabricated on the same chip. The low power consumption of FG sensors and their small area allows designing ultra-low power low-cost UV dose measurement edge nodes for smart sterilization systems, comprising UVC irradiation sources and sensors. The edge nodes provide the feedback on the efficiency of the performed disinfection and can be reprogrammed up to 10,000 times. Additionally, the feasibility of developed FG sensors for UVC imaging was demonstrated. For this purpose, individual sensors were integrated into 256 × 256 pixel matrices.

## Figures and Tables

**Figure 1 sensors-23-02509-f001:**
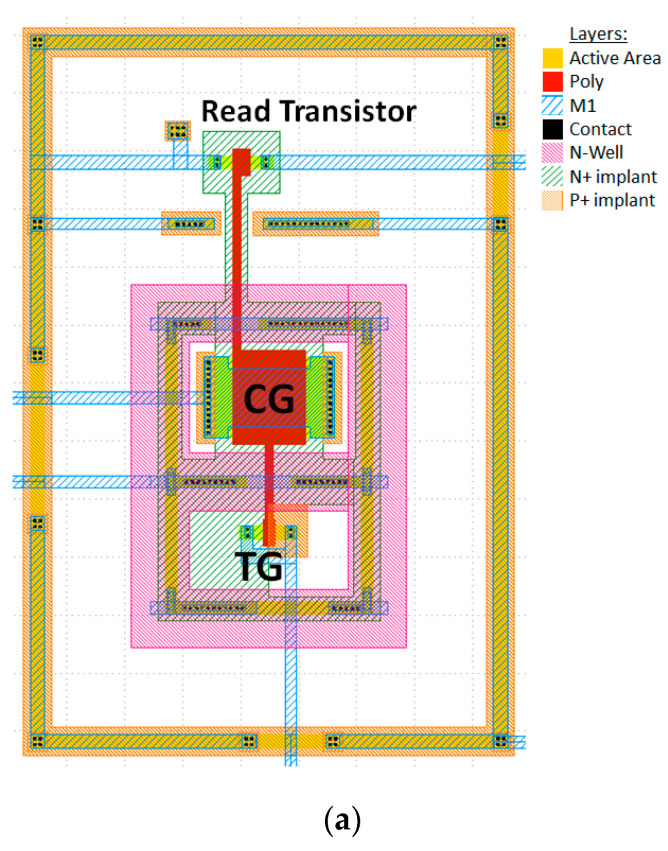
(**a**) C-Flash NVM cell layout. (**b**) C-Flash NVM cell cross-section.

**Figure 2 sensors-23-02509-f002:**
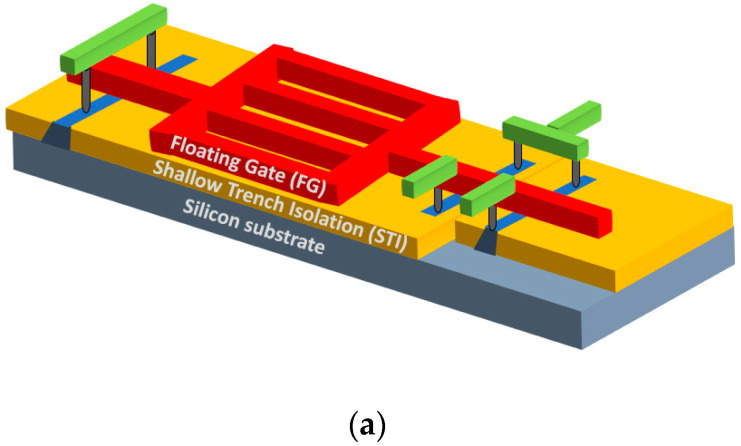
(**a**) Three-dimensional (3D) illustration of “grilled” C-Sensor. (**b**) Schematic illustration of the C-Sensor and discharge of Poly FG. (**c**) Band diagram illustrating FG discharge by UVC.

**Figure 3 sensors-23-02509-f003:**
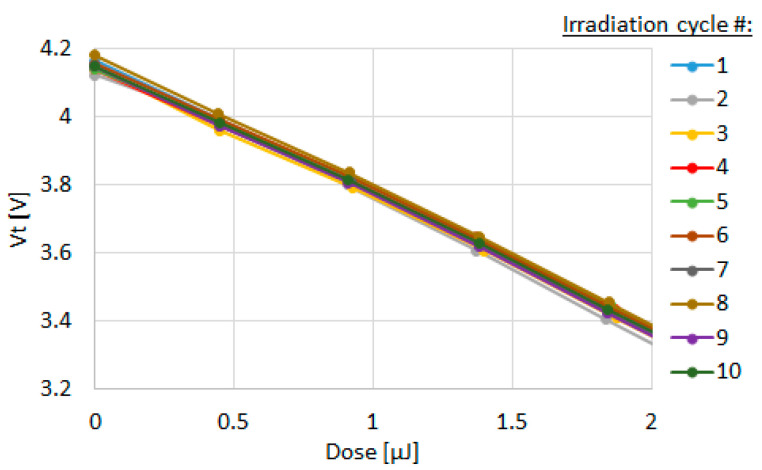
Kinetics of FG discharge after 10 program-discharge cycles. Degradation was not observed.

**Figure 4 sensors-23-02509-f004:**
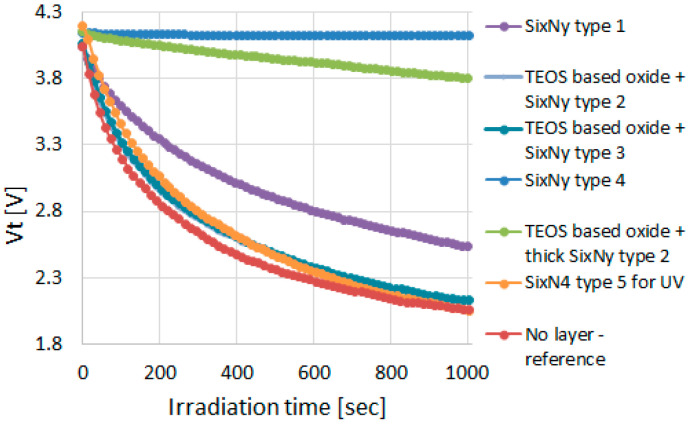
Y-Flash cells Vt change as a result of 255 nm UV irradiation. The effect of various nitride containing layers is evaluated.

**Figure 5 sensors-23-02509-f005:**
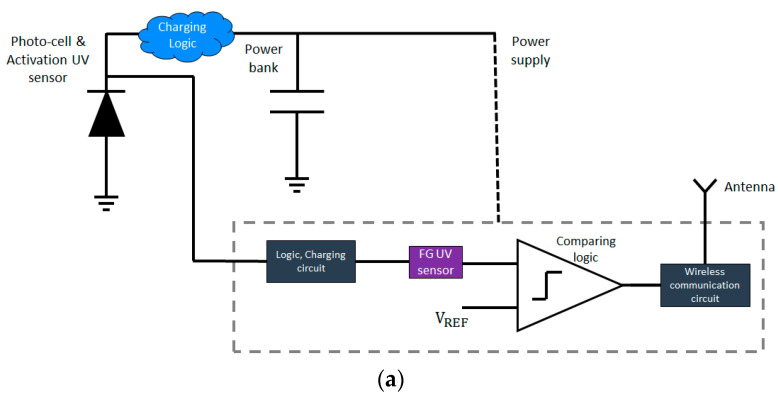
(**a**) UV sensor edge unit schematic (integrated FG UV sensor) (**b**) Disinfection system with multiple edge units, (illustration).

**Figure 6 sensors-23-02509-f006:**
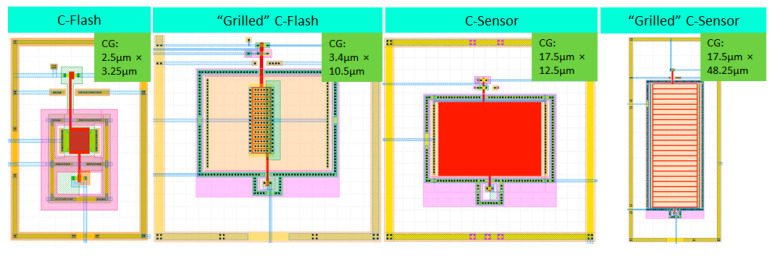
Types of FG-based C-Flash and C-Sensor devices.

**Figure 7 sensors-23-02509-f007:**
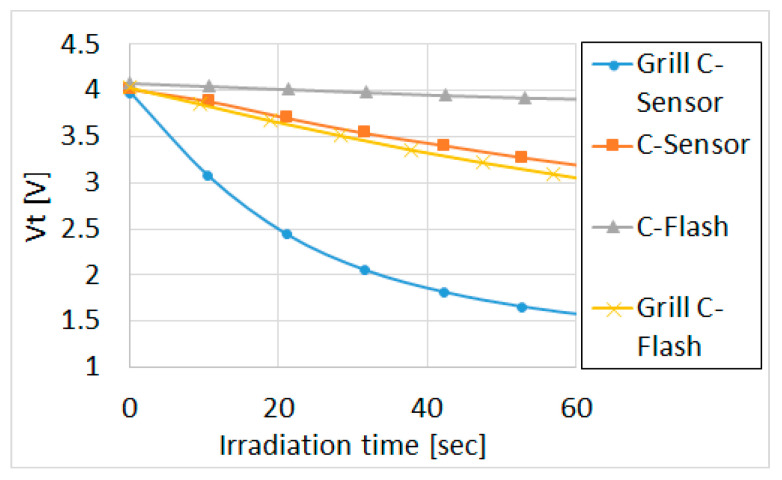
Different FG sensors discharge rate by 255 nm UV, with intensity of ~500 µW/cm².

**Figure 8 sensors-23-02509-f008:**
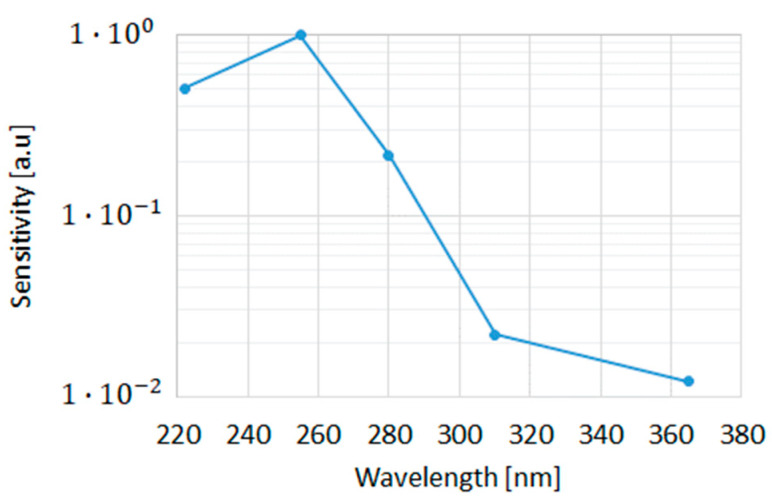
Grill C-Sensor normalized sensitivity for different UV wavelengths.

**Figure 9 sensors-23-02509-f009:**
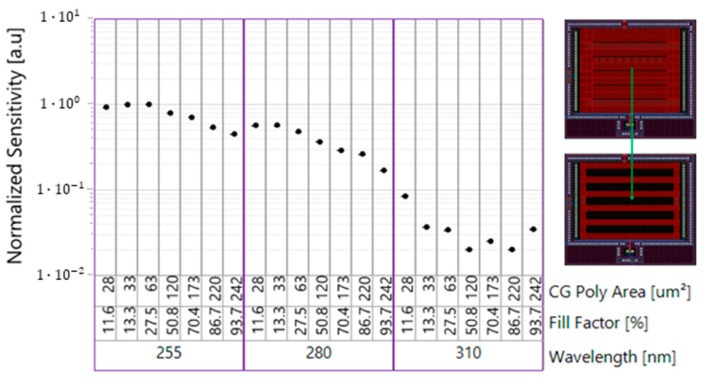
Grill C-Sensor normalized sensitivity for different UV wavelengths vs. fill factor.

**Figure 10 sensors-23-02509-f010:**
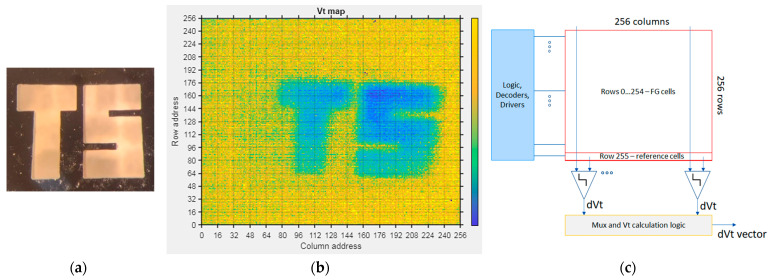
(**a**) Si template used for UV irradiation. (**b**) FG UV sensor matrix after exposure to 280 nm UV. (**c**) Schematic diagram of the read-out of FG sensor array.

**Table 1 sensors-23-02509-t001:** Evaluated Layers at Back End.

Layer Name	Type	Vt Decay after 1000 s Irradiation (V)
Interdielectric layer 1	SixNy type 1	1.503
Interdielectric layer 2	TEOS based oxide + SixNy type 2	1.958
Interdielectric layer 3	TEOS based oxide + SixNy type 3	1.929
Passivation layer 1	SixNy type 4	0.025
Passivation layer 2	TEOS based oxide + thick SixNy type 2	0.347
Passivation layer 3	SixN4 type 5 for UV	2.146
No Layer	No layer- reference	1.982

**Table 2 sensors-23-02509-t002:** UV sensor types.

	Type	Bulk Si Diode	SOI Diode	SiC, GaN Diode	FG Device
Parameter					
CMOS integrated		Yes	Yes	No	Yes
Power consumption		Medium	Medium	Medium	Low
Degradation		Low	High	Low	Low
Sensitivity		High	High	High	High
Response time		Low	Low	Low	High
Cost		Medium	High	High	Low

## Data Availability

Raw data related to the measurements of developed devices are available. Specimens of the fabricated UV sensors, as well as demonstration kits can be provided to interested parties under individual agreements.

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
