# Peer review of "High-Sensitivity CMOS-Integrated Floating Gate-Based UVC Sensors"

_sensors, 2023, doi:10.3390/s23052509_

Round 1
Reviewer 1 Report
The manuscript by Yampolsky et al. reported UVC sensors based on floating gates. They have shown the smooth integration of the floating gate-based device in the CMOS system for an efficient UVC sensing system for practical applications. The UVC sensing systems are very important for various practical applications, particularly, to check/sense the UVC dose. The reported results are significant and should be published in the journal ‘Sensors’. However, the article lacks clarifications in several places. The manuscript can be accepted after providing clarifications/minor revisions of the issues raised below.
(1) The authors have mentioned ‘… FG sensors are solar blind, due to the energy barrier for electrons between the FG and the surrounding silicon dioxide…’ This is not completely clear to the readers. The authors should add 1-2 sentences and/or band diagram to explain the reason.
(2) On page no. 3, the authors have mentioned ‘…The photoionized electrons which are transferred into the surrounding oxide (and then to the substrate) are generated at the edges of Poly electrodes only…’ What do the authors mean by poly electrodes? The author should explain clearly. The schematic diagrams are also not giving a complete idea regarding it.
(3) On page no. 3, the authors have mentioned ‘… The device can be charged and discharged repeatedly up to 10k times and has no significant sensitivity degradation after cycling and exposure to UV…’ The authors should have presented the charging-discharging diagram to give an idea of its degradation properties and then can be compared with ref. 16.
(4) In fig. 6, the authors have shown the discharge rate of the UV sensors. The authors should have done the discharge curve fitting and reported the exact discharge time and compare it with other results to show the superiority of their devices.
(5) The authors mentioned ‘..Smaller fill factor resulted in the maximum sensitivity at all UV wavelengths..’, but did not give any suitable explanation behind it. The authors should clarify it.
(6) The authors should give a complete schematic diagram for their UV imaging system. They have mentioned the sensor part, but have ignored the read-out part based on CMOS. It will be much easier for the readers to understand with a complete diagram.
(7) The authors should give a table for comparison of their sensor performance with other contemporary sensors working in a similar wavelength range.
Author Response
Dear Reviewer, thank you for the detailed inputs and valuable recommendations. We have made adjustments in the paper accordingly. Please see our reply to your comments below:
- The discharge of FG happens mainly by excitation of electrons from the valance band of the charged Poly floating gate over the energy barrier at the Poly-SiO2 interface. The barrier is ~4.3 eV. Energy of 4.3 eV corresponds to the wavelength of ~288 nm. This makes the FG sensor solar blind, i.e., not sensitive to wavelengths in the visible range. We added band diagram (Figure 2c) for illustration.
- The FG made of ~2500 Å thick Polysilicon (Poly) absorbs UVC photons (absorption depth <100 Å), so that most of the generated photoelectrons do not reach Poly-SiO2 surface. The photoionized electrons which are transferred over the energy barrier into the silicon dioxide (and then to the substrate) are generated only at the edges of Polysilicon floating gate adjacent to the shallow trench isolation (STI). We added illustration of the photon absorption in Figure 2b.
- We added Figure 3 to show the results of multiple charging-discharging cycles. 10k cycles stands for the amount of program/erase cycles that the used in sensor design NVM technology was qualified for.
- EPROM devices featuring the same discharge mechanism and using UV transparent back end, but different geometry of the FG and thickness of oxide under it, are typically discharged in ~20 minutes (see graph for C-Flash NVM at Figure 7). This is ~50 times slower than for the developed UVC sensors. This comparison is included into the text of the paper.
- In continuation to the explanation in (2), the effect relates to lower total capacitance of the FG and the increase of the FG periphery length.
- We added a schematic diagram (Figure 10c) of the read-out. Read-out of the sensor array is done by ramping gate (CG) voltage on rows of FG cells and comparing them to rows of reference cells (NMOS transistors) with reference voltage at gate.
- We added Table 2, comparing different types of UVC sensors. The advantages of our FG devices over other types of UVC sensors are summarized in Table 2. FG devices show excellent sensitivity, low power consumption, low cost and low degradation. The major limitation relates to the response time. The developed low cost integrated into CMOS sensors are recommended for applications that require measuring of total doses of radiation.
Reviewer 2 Report
Good and organized technical report to the UV sensor society.
For better understanding,
1) In the title, 'MOS' or 'CMOS compatible' term is desirable to make floating gate clearer.
2) In Figure 1, a cross-sectional view will be good for readers' understanding. In explaing Figure 7.
3) They need to explain the reason of the solar-blindness of the sensor fabricated.
Author Response
Dear Reviewer, thank you for your valuable inputs. We have made adjustments in the paper accordingly. Please see our reply to your comments below:
- Thank you! We changed the title of paper according to your recommendation.
- We added the cross-section of the device (Figure 1b).
- The discharge of FG happens mainly by excitation of electrons from the valance band of the charged Poly floating gate over the barrier at the Poly-SiO2 interface. The barrier is ~4.3 eV. Energy of 4.3 eV corresponds to the wavelength of ~288 nm. This makes the FG sensor solar blind, i.e., not sensitive to wavelengths in the visible range.
We added the band diagram (Figure 2c) for illustration.